# Non-Coding RNA Regulates the Myogenesis of Skeletal Muscle Satellite Cells, Injury Repair and Diseases

**DOI:** 10.3390/cells8090988

**Published:** 2019-08-27

**Authors:** Yue Zhao, Mingming Chen, Di Lian, Yan Li, Yao Li, Jiahao Wang, Shoulong Deng, Kun Yu, Zhengxing Lian

**Affiliations:** 1Beijing Key Laboratory for Animal Genetic Improvement, National Engineering Laboratory for Animal Breeding, Key Laboratory of Animal Genetics and Breeding of the Ministry of Agriculture, College of Biological Sciences, College of Animal Science and Technology, China Agricultural University, Beijing 100193, China; 2CAS Key Laboratory of Genome Sciences and Information, Beijing Institute of Genomics, Chinese Academy of Sciences, 100101 Beijing, China

**Keywords:** non-coding RNAs, skeletal muscle satellite cells, myogenesis, muscle diseases

## Abstract

Skeletal muscle myogenesis and injury-induced muscle regeneration contribute to muscle formation and maintenance. As myogenic stem cells, skeletal muscle satellite cells have the ability to proliferate, differentiate and self-renew, and are involved in muscle formation and muscle injury repair. Accumulating evidence suggests that non-coding RNAs (ncRNAs), including microRNAs (miRNAs), long non-coding RNAs (lncRNAs) and circular RNAs (circRNAs), are widely involved in the regulation of gene expression during skeletal muscle myogenesis, and their abnormal expression is associated with a variety of muscle diseases. From the perspective of the molecular mechanism and mode of action of ncRNAs in myogenesis, this review aims to summarize the role of ncRNAs in skeletal muscle satellite cells’ myogenic differentiation and in muscle disease, and systematically analyze the mechanism of ncRNAs in skeletal muscle development. This work will systematically summarize the role of ncRNAs in myogenesis and provide reference targets for the treatment of various muscle diseases, such as muscle dystrophy, atrophy and aberrant hypertrophy.

## 1. Introduction

As one of the vertebrate striated muscles, skeletal muscle is constituted of myofibers, which have giant multinuclear cells. By partially reproducing embryonic development processes, adult skeletal muscle is endowed with regenerative potential, which mainly relies on skeletal muscle satellite cells [1]. As the stem cells of skeletal muscle, skeletal muscle satellite cells are located between the skeletal muscle myofiber membrane and the basal lamina membrane [2,3,4]. Under normal physiological conditions, satellite cells are in a quiescent state. In the context of physiological stimuli (physical exercise, acute muscle injury or pathological conditions), quiescent satellite cells are activated and give rise to myogenic progenitors, which then proliferate, migrate, align, differentiate and fuse to form new multinucleated myotubes and restore tissue functionality [5,6,7,8]. Besides, a part of the activated satellite cells is returned to quiescence to replenish the stem cell pool by self-renewing [1,9]. These characteristics of satellite cells are a prerequisite for muscle formation and maintenance. The enormous myogenic potential of skeletal muscle satellite cells is attributed to the expression of paired box families (Pax3 and Pax7), myocyte enhancer factor 2 (MEF2) family proteins and many myogenic regulatory factors, such as MyoD, Myf5, Myogenin and MRF4 [10,11].

Accumulating evidence suggests that non-coding RNAs (ncRNAs), including microRNAs (miRNAs), long non-coding RNAs (lncRNAs) and circular RNAs (circRNAs), are widely involved in a series of subsequent myogenesis processes such as satellite cell activation, proliferation, differentiation and self-renewal [12,13,14,15] (Table 1 and Table 2). These ncRNAs are emerging as integral components of the gene regulatory networks in a variety of biological processes. miRNAs are a class of endogenous, conserved small non-coding RNAs (~18 to 22 nt), which are transcribed by RNA polymerase II [16,17]. Functionally, miRNAs negatively regulate the expression of their target genes at the post-transcriptional level by inducing target mRNAs’ degradation and/or repressing translation [18,19]. Due to the fact that they are misregulated in pathophysiological conditions, miRNAs are often regarded as biomarkers and potential therapeutic targets [20]. With weak or no protein-coding potential, lncRNAs are a class of RNA transcripts (>200 nt), which have poor evolutionary conservation compared with miRNAs [21]. LncRNAs have complex spatial structures and diverse functions. However, in summary, lncRNAs play a crucial role at three levels. At the epigenetic level, lncRNAs mediate chromatin remodeling and modification to exert an epigenetic regulatory role [22,23]. Furthermore, lncRNAs can interact with transcription factors to affect transcriptional regulation [24,25]. In addition, lncRNAs can also bind to mRNAs to form double strands, which specifically regulate the various post-transcriptional processes of mRNAs, including splicing, transport, translation and degradation [26,27]. CircRNAs are another class of endogenously expressed non-coding RNAs with a closed continuous loop in which the 5’ and 3’ ends are joined together [28,29,30]. Compared with miRNAs and lncRNAs, circRNAs have greater stability and higher conservation in mammalian cells [30]. While circRNAs were discovered decades ago, their function is only now beginning to be understood. Up to now, it has been discovered that circRNAs regulate gene expression by acting as a sponge for miRNAs and RNA-binding proteins to play a role as competing endogenous RNAs (ceRNAs), and circRNAs can also be translated into proteins [31,32,33,34]. Further study has demonstrated that circRNAs also are involved in diseases, suggesting that circRNAs could be used as a potential target for diagnosis and therapeutics [35,36].

An increasing number of studies have identified many ncRNAs and shed light on their mechanisms. However, certain mysterious functions of these ncRNAs have not been completely unveiled, especially their functions in skeletal muscle myogenesis. Here, we will discuss the current understanding of the molecular mechanisms and mode of action through which ncRNAs regulate satellite cells’ function. This review aims to expand the understanding of skeletal muscle biology and provide reference targets for the treatment of various muscle diseases, such as muscle dystrophy, atrophy, aberrant hypertrophy and cachexia.

## 2. ncRNAs and Myogenesis

Many ncRNAs can not only maintain the static state of skeletal muscle satellite cells, but also participate in the processes of proliferation, differentiation and self-renewal. However, the regulatory effects of different ncRNAs are different. Here, we will systematically review the ncRNAs that play a role in multiple processes and those that are important in specific processes. The transcription factor paired box-protein-7 (Pax7) and the transcription factor paired box-protein-3 (Pax3) are essential for the establishment and maintenance of adult satellite cells’ status [37,38,39]. Studies have shown that transcription factors Pax3 and Pax7 are regulated by miR-1 and miR-206 at the post-transcriptional level. In quiescent satellite cells, an experiment in mice without TNF receptor-associated factor 6 (TRAF6) suggested that the expression of miR-1 and miR-206 is repressed by TRAF6/c-JUN signaling, which increases the expression of Pax7 and further maintains satellite cells’ quiescence [40]. During satellite cells’ differentiation, miR-1 and miR-206, which are involved in the post-transcriptional regulation of Pax7, are sharply up-regulated, while they are strikingly decreased during skeletal muscle regeneration [41]. This suggests that miR-1 and miR-206 restrict the proliferative potential of satellite cells and promote cell differentiation. However, another study demonstrated that there was no significant difference in the fraction of satellite cells between WT and miR-206 KO mice at 3 days after cardiotoxin (CTX) injury [42]. In addition, histone deacetylase 4 (HDAC4) is considered as an inhibitor to repress cell differentiation, which plays its role by inhibiting members of the myocyte enhancer factor-2 (MEF2) transcriptional activity to suppress the myogenic regulators Myf5, MRF4 and Myogenin [43]. Here, miR-1 and miR-206 can suppress the post-transcriptional expression of HDAC4 to promote satellite cells’ differentiation and muscle development in vivo [44,45]. In a hypoxic environment, down-regulating the expression of miR-1 and miR-206 can activate the Notch signaling pathway, further up-regulate the expression of Pax7, participate in the regulation of satellite cell proliferation and promote the self-renewal of satellite cells [46]. Therefore, miR-1 and miR-206 participate in the whole process of satellite cell development and are indispensable micoRNAs for regulating satellite cell development. It has been verified that MiR-27b directly targets the 3’-UTR of Pax3 mRNA. Further, its interference with miR-27b function results in the continuation of Pax3 expression, leading to more proliferation and a delay in the onset of differentiation. The introduction of miR-27b antagomir at a site of muscle injury affects Pax3 expression and regeneration in vivo [47]. In recent studies on satellite cells in goats, miR-27b was also found to target the 3’-UTR of Pax3, and the expression of miR-27b decreased gradually during the proliferation of satellite cells but increased during cell differentiation [48]. In addition, a recent study showed that miR-27b also inhibits the proliferation and promotes the differentiation of pig muscle satellite cells by targeting the MyoD family inhibitor (MDFI) [49].

LncRNA H19 was one of the earliest lncRNAs to be identified as being highly expressed in animal skeletal muscle [50]. It has been reported that the imprinted gene encoding lncRNA H19 is highly expressed in satellite cells, and the satellite cells decreased by 50% in adult muscle of H19-deleted mice [51]. This evidence suggests that lncRNA H19 might be involved in the maintenance of quiescent satellite cell pools. A previous study showed that maternal-specific H19-DMR deletion results in the activation of the IGF2-IGF1R pathway, activating hematopoietic stem cells [52]. Interestingly, the IGF2 transcript levels were increased in H19-deleted adult tibialis muscle, but IGF1R was not regulated [51]. Whether lncRNA H19 protects the skeletal muscle satellite cells from activation through IGF2-IGF1R signaling remains a mysterious question. During both human and mouse satellite cell differentiation, H19 was up-regulated, and down-regulated H19 suppressed satellite cells’ differentiation by reducing the mRNA level of myogenin and MyHC. In the same way, H19 knockout mouse satellite cells decreased differentiation and displayed abnormal skeletal muscle regeneration after injury. However, this injury was treated to some extent by the reintroduction of miR-675-3p and miR-675-5p, which target Smad1, Smad5 and Cdc6 [53]. In addition, H19 can also promote satellite cells’ differentiation by suppressing the myoblast inhibitory genes Sirt1 and FoxO1. In detail, H19 induced the down-regulation of Sirtuin 1 (Sirt1), and Forkhead box class O transcription factor-1 (FoxO1) can increase the expression levels of MyoG and MyHC [54].

Except for ncRNAs that play a regulatory role in a variety of processes, many ncRNAs play important roles in the processes of quiescence and activation. Further research reveals that miR-489 maintains satellite cells’ quiescence by targeting and repressing oncogene Dek, the protein product of which alternatively localizes to the more differentiated daughter cells [55]. Similarly, the cell cycle exit of quiescent satellite cells is also regulated by miR-195 and miR-497. In detail, miR-195 and miR-497 target and inhibit the expression of cyclin D2 (Ccnd2) and cell division cycle 25a/b (Cdc25a/b), then further induce cell cycle withdrawal to maintain quiescence [56]. In addition, the lncRNA Uc.283+A controls the processing of pri-miR-195 by complementing with the immature pri-miR-195 and preventing its cleavage by Drosha, which is required for the formation of functionally mature miR-195 [57]. While the mechanism of the lncRNA Uc.283+A in the quiescence and activation of skeletal muscle satellite cells has not yet been reported, the lncRNA Uc.283+A can rapidly destroy the functional miR-195 by blocking its formation, which is needed to maintain satellite cells’ quiescence, so it may be a potentially key regulator. Recent research has shown that Notch induces the transcription of the quiescence-specific mirtron miR-708, which represses Tensin3 to inhibit the activation of focal adhesion kinase (FAK), which further stabilizes satellite cells within their niche [58]. This suggests that Notch signaling can regulate the transition between quiescence and activation of satellite cells via the Notch-miR-708-Tensin3 axis (Figure 1). Regarding the mechanisms, miR-31 can bind to Myf5 and form messenger ribonucleoprotein (mRNP) granules, which inhibit the translation of Myf5, so that satellite cells can be maintained in quiescence. After the activation of satellite cells, the down-regulated miR-31 leads to the dissociation of mRNP granules, which rapidly releases Myf5 mRNA and promotes the translation and accumulation of Myf5 protein [59]. TRAF6 facilitates the expression of Pax7 in satellite cells by inhibiting the expression of mir-1 and mir-206. Meanwhile, the lncRNA 1/2-sbsRNA(B2) may target TRAF6 mRNA and reduce the half-life of TRAF6 mRNA, which further represses the protein expression of TRAF6 in C2C12 cells [60].

In addition, ncRNAs that regulate proliferation and differentiation should not be ignored. Exposing inhibited satellite cells to the stimulatory Wnt signaling pathway restores their proliferation rate [61]. Recently, it has been reported that the Gtl2-Dio3 miRNA mega-cluster regulated by MEF2A inhibits the myoblast Wnt signaling pathway [62], suggesting that the Gtl2-Dio3-miRNA axis is a potential guarder for the proliferation of satellite cells, though this process occur in myoblasts. In addition, miR-133b is a key component of the canonical Wnt pathway and is an inhibitor of Pax7 expression; it acts via a site adjacent to the miR 206 binding site in the Pax7 3 UTR, which allows differentiation to proceed by relieving Pax7-mediated repression of the myogenic program [63]. Linc-MD1 is localized in the cytoplasm, is polyadenylated and acts as a natural decoy for miRNAs to be involved in the timing of muscle differentiation [64]. After the differentiation of mouse satellite cells, linc-MD1 is activated and then the expression of transcription factors mastermind-like 1 (MAML1) and MEF2C, which activate muscle-specific gene expression, is regulated by sponging miR-133 and miR-135 [64]. In primary bovine skeletal muscle-derived satellite cells, miR-17 can up-regulate the transcription of MYH3, MyoD1 and MyoG to accelerate the differentiation process. Bovine skeletal muscle satellite cells which over-express with miR-17 and miR-19 can differentiate more myotubes and have a higher fusion efficiency than miR-17 alone, suggesting that miR-17 and miR-19 jointly promote skeletal muscle cell differentiation [65]. Similarly, miR-139 can target DHFR mRNA to inhibit satellite cell differentiation in bovine skeletal muscle satellite cells [66]. Besides, miR-34c inhibits the proliferation and promotes the differentiation of porcine muscle satellite cells by inhibiting Notch1 expression [67].

**Table 1 cells-08-00988-t001:** microRNAs (miRNAs) and targets in the myogenesis of skeletal muscle satellite cells.

miRNA	Function	Target	Reference
miR-1	Maintains cell quiescence Promotes satellite cell self-renewal Promotes cell differentiation	Pax7 HDAC4	[37,38,39,40,41,42]
miR-206	Maintains cell quiescence and satellite cell self-renewal Promotes cell differentiation	Pax7 HDAC4	[37,38,39,40,41,42]
miR-27	Activates satellite cells	Pax3	[47]
miR-27b	Promotes cell differentiation	MDFI	[49]
miR-31	Maintains cell quiescence	Myf5	[59]
miR-489	Maintains cell quiescence Regulates satellite cell self-renewal	Dek	[55]
miR-195	Maintains cell quiescence	Ccnd2Cdc25a/b	[56]
miR-497	Maintains cell quiescence	Ccnd2Cdc25a/b	[56]
miR-708	Activates satellite cells Regulates satellite cell self-renewal	Tensin3	[58]
miR-17/miR-19	Promotes cell differentiation	MRFs	[65]
miR-139	Inhibits cell differentiation	DHFR	[66]
miR-34c	Inhibits cell proliferation Promotes cell differentiation	Notch1	[67]
miR-133b	Promotes cell differentiation	Pax7	[63]

LncMyoD is directly activated by MyoD during myoblast differentiation, and has been proved to have a positive regulatory effect on myogenic differentiation. Regarding its mechanism, lncMyoD directly binds to IGF2-mRNA-binding protein 2 (IMP2) and results in decreased translation levels of proliferation genes such as N-Ras and c-Myc [68]. Lnc-mg has been shown to be a molecular sponge of miR-125b in vitro, and insulin-like growth factor 2 (IGF2) is a direct target of miR-125b in muscle stem cells. Therefore, lnc-mg is a key myogenesis enhancer as a miR-125b ceRNA that regulates the abundance of IGF2 protein [69]. Myogenic differentiation is also regulated by linc-YY1 in muscle satellite cells. Linc-YY1, which is also induced by MyoD, interacts with YY1 protein to expel YY1/PRC2 from the target promoter, thereby activating gene expression to promote myogenic differentiation and muscle regeneration [70]. Furthermore, the treatment of regenerating muscles with si-linc-YY1 led to decreases in Pax7, MyoD, Myogenin and e-MyHC [70]. It can be determined that the process of muscle regeneration is affected by the increase or decrease in the linc-YY1 function in the injured muscle.

## 3. ncRNAs, Muscle Injury Repair and Diseases

The activation, proliferation, differentiation and self-renewal of skeletal muscle satellite cells are closely related to biological processes such as skeletal muscle growth and development, regeneration, injury repair, muscular dystrophy, muscular atrophy and aberrant muscle hypertrophy, and the disorder of any link can lead to the occurrence of muscle diseases. Undoubtedly, a large number of ncRNAs are involved in the biological processes of the above-mentioned skeletal muscle satellite cells, which play a crucial role in the occurrence and development of muscle diseases. This suggests that ncRNAs may serve as a potential therapeutic target for the treatment of skeletal muscle disease. Here, we mainly review the role of ncRNAs in muscle injury repair and diseases (Figure 2).

### 3.1. ncRNAs in Muscle Dystrophy

Muscle dystrophy is a genetic disease; Duchenne muscular dystrophy (DMD), caused by the mutation of the dystrophin gene on the X chromosome, is the most common and severe type of muscle dystrophy.

As a structural protein, dystrophin can link the cytoskeleton and form a large membrane-associated multi-protein complex (dystrophin-associated protein complex, DAPC) to stabilize the sarcolemma [71]. In DMD, the absence of dystrophin at the sarcolemma delocalizes and down-regulates nitric oxide synthase (nNOS), which alters the status of the nitrosilation of Histone Deacetilases (HDACs) and its chromatin association [43,72]. The loss of dystrophin in DMD patients and mdx mice leads to a decrease in DAPC and a corresponding injury of NO production [73]. Cacchiarelli et al. found that the expression of a specific subset of miRNAs was deregulated by the differential HDAC2 nitrosylation state in Duchenne and wild-type conditions [74]. Further study demonstrated that the activation of miR-1 and miR-29 was closely linked to HDAC2 release from their respective promoters in humans and mice [74].

It has been reported that the myomiRs (miR-1, miR-206 and miR-133) are highly expressed in the serum of DMD patients [75,76]. Furthermore, their levels were correlated with the severity of DMD disease, suggesting that myomiRs are novel biomarkers for the diagnosis of DMD [77]. The deletion of miR-206 in mice delayed the muscle regeneration induced by cardiotoxin injury, suggesting that miR-206 can promote skeletal muscle regeneration and is involved in muscle injury repair [42]. Interestingly, the absence of miR-206 accelerated and aggravated the dystrophic phenotype in a DMD mouse model by weakening the suppression by miR-206 of myogenesis inhibitors Pax7, Notch3 and IGFBP5 [42]. Compared with other myomiRs, the expression level of miR-206 is elevated in distrophic mdx muscle due to the fact that it activates skeletal muscle satellite cells’ differentiation by inhibiting the expression of Pax7 and HDAC4 [74]. Furthermore, miR-206 can target Utrophin, a systrophin protein homolog involved in a compensatory mechanism in DMD pathology [78]. Using miR-127 transgenic model mice, Zhai et al. found that miR-127 can promote satellite cells’ differentiation to accelerate muscle regeneration by targeting the Sphingosine 1 Phosphate Receptor 3 (S1PR3) gene. Further study indicated that the over-expression of miR-127 can significantly improve the disease phenotype of muscular dystrophy model mdx mice [79]. A recent study demonstrated that miR-200c plays an important role in skeletal muscle regeneration in mdx mice with muscular dystrophy. In detail, miR-200c might increase reactive oxygen species (ROS) production and induce the phosphorylation of p66Shc in Ser-36 to cause muscle dystrophy and wasting [80]. The latest research showed that the synthetic preimplantation factor (sPIF) can promote DMD myoblasts’ differentiation, increase the expression of utrophin, and reduce muscle fibrosis, possibly via the up-regulation of miR-675 and inhibition of miR-21 expression [81]. In addition, the lncRNA linc-MD1 is expressed in newly regenerating fibers and is abundant in dystrophic conditions, suggesting that linc-MD1 might also be involved in the regulation of muscle dystrophy [64]. Recently, RNA sequencing demonstrated that the expression of circRNAs had a unique feature between normal and dystrophic human myoblasts derived from DMD patients. In the differentiation of two types of myoblasts, circ-QKI and circ-BNC2 were up-regulated in normal myoblasts, but were down-regulated in DMD conditions [82]. Overall, this evidence suggests that lncRNAs and circRNAs may play a functional role in DMD.

Myotonic dystrophy (DM) is a progressive muscular dystrophy with two types, i.e., type-1 and type-2. DM1 is a multisystemic disorder disease caused by abnormal mRNA splicing, which is induced by the expanded CTG repeat in dystrophic myotonic protein kinase (DMPK), while DM2 is caused by CCTG repeat expansion in the first intron of the CCHC-type zinc finger nucleic acid binding protein (CNBP) [83]. An analysis of DM1 biopsies obtained from 15 patients showed that the expression of miR-1 and miR-335 were up-regulated, whereas the expression of miR-29b, miR-29c and miR-33 were down-regulated compared with normal muscles [84,85]. It is reported that the alternative splicing factor muscle blind-like (MBNL) was down-regulated by miR-23b and miR-218, which further accelerated the pathogenic misplacing events in the myoblasts with the neuromuscular disease myotonic dystrophy type-1 [86]. A recent study showed that the expression of circRNA (RTN4_03 and ZNF609) was increased in differentiated myogenic cell lines derived from DM1 patients, suggesting the crucial role of circRNA in DM1 patients [87].

### 3.2. ncRNAs in Muscle Atrophy

With the typical symptom of muscle quality loss, muscle atrophy is a disease caused by increased protein degradation or decreased protein synthesis in skeletal muscle [88]. Evidence from numerous studies indicates that ncRNAs play an important role in the regulation of muscle atrophy. Muscle atrophy can be divided into primary and secondary muscular disease. Primary muscle atrophy is directly caused by muscle diseases such as DMD and DM1 [75,89]. Meanwhile, secondary muscular disease is usually a complication of other diseases [76,90].

The processing of miRNAs from precursor to mature and their exportation to the nucleus depends on many proteins, such as the DROSHA/DGCR8 complex, Exportin-5 and Dicer enzyme [91]. These proteins involved in miRNAs biogenesis and production have been shown to be important in regulating muscle atrophy. The loss of Dicer activity during embryogenesis reduced muscle-specific miRNAs, which further resulted in decreased muscle mass and abnormal myofiber morphology [92]. In addition, miRNA machinery protein Argonaute2 (Ago2) is also important for skeletal muscle atrophy [93]. After the endonuclease activity of Ago2 was significantly repressed by the loss of Crystallin-B in mice, the body weight and myofiber cross-sectional area were obviously reduced [93]. This evidence further indicates the relationship between miRNAs and muscle atrophy. The serum levels of muscle-specific miRNAs, such as miR-1, miR-23a, miR-206, miR-208b and miR-499, were all significantly increased after hindlimb unloading for seven days in mice and was able to induce severe muscle atrophy [94]. Furthermore, the expression levels of miR-23a, miR-206 and miR-499 were positively correlated with the ratio of soleus volume loss [94], suggesting that circulating miR-23a, miR-206 and miR-499 might be used as candidate biomarkers for the diagnosis of muscle atrophy. In multiple models of skeletal muscle atrophy, the E3-ubiquitin ligases Atrogin and muscle-specific RING finger protein 1 (MuRF1) are crucial for accelerating the degradation of muscle sarcomeric proteins [95]. miR-23a can target Atrogin-1 and MuRF1 and inhibit their translation, and the ectopic expression of miR-23a protects muscles from atrophy in vitro and in vivo, indicating that miR-23a is a critical regulator in muscular atrophy. Moreover, FoxO1 can regulate the expression of E3-ubiquitin ligases Atrogin-1/MAFbx and MuRF1, and miR-486 inactivates atrophy signaling in skeletal muscle by coordinately inhibiting its targets pentaerythritol tetranitrate (PTEN) and FoxO1 [96]. Furthermore, up-regulated miR-1 also promoted skeletal muscle atrophy through targeting HSP70, which led to a decreased phosphorylation of AKT, an elevated activation of FoxO3 and the up-regulation of MuRF1 and Atrogin-1 [97]. Li et al. found that miR-29b was elevated in multiple in vivo and in vitro atrophy models, that the over-expression of miR-29b contributes to muscle atrophy by targeting IGF-1 and PI3K (p85α) and that its inhibition attenuates muscle atrophy [98]. As mentioned above, Notch signaling can regulate the maintenance of quiescent satellite cells and their differentiation. Here, miR-199b targets JAGGED1 (JAG1) to activate the Notch1 signaling pathway and further promotes the proliferation of porcine muscle satellite cells, suggesting that miR-199b is a potential therapeutic target for muscle atrophy [99].

Amyotrophic lateral sclerosis (ALS) is a heterogeneous neurodegenerative disease. MiR-206 was increased in the muscles of four early-stage ALS patients, and was considered as a potential biomarker of ALS [100,101]. In mice, the loss of miR-206 accelerated the ALS process and induced severe skeletal muscle atrophy by targeting HDAC4 [102]. A miRNA profiling analysis demonstrated that a number of differentially expressed miRNAs were identified in ALS patients and healthy controls. Compared with control, the expression levels of miR-206 and miR-143-3p were increased and miR-374b-5p was decreased [101]. This suggested that miRNAs also play an important role in ALS. Spinal muscular atrophy (SMA) is an autosomal recessive neuromuscular disease, which is caused by deletions or mutations in the survival motor neuron (SMN1) gene [103]. The lncRNA SMN-AS1 is transcribed from the antisense strand of SMN and is highly enriched in neurons [104]. SMN-AS1 accelerates muscle atrophy via recruiting PRC2 to the SMN promoter and transcriptionally repressing SMN expression [105]. In addition, the selective disruption of SMN-AS1-mediated PRC2 recruitment resulted in active SMN and better SMA phenotypes in mice [105].

Single cell analysis revealed that the lncRNA Pvt1, which is activated in early muscle atrophy, is involved in the regulation of muscle atrophy and the mitochondrial network. In detail, the down-regulation of the lncRNA Pvt1 results in the destabilization of c-Myc and mediates the increase in Bcl-2, which is a central node in the regulation of apoptosis and atrophy [106]. A transcriptomic analysis between hypertrophic broilers and leaner broilers demonstrated that a novel lncRNA lncIRS1 sponges miR-15a/miR-15b-5p/miR-15c-5p and activates the IGF1-PI3K/Akt signaling pathway to control muscle atrophy [107]. In addition, mechanical unloading induces obvious skeletal muscle atrophy. A study proved that lncMUMA is a muscle-enriched lncRNA, which decreased most in hindlimb suspension mice [108]. Further research revealed that lncMUMA, as a sponge of miR-762, can promote myogenic differentiation, regulate the core myogenic regulator MyoD in vitro and reverse the established muscle atrophy in hindlimb suspension mice to prevent muscle atrophy development [108].

### 3.3. ncRNAs in Aberrant Muscle Hypertrophy

Healthy muscles enlarge through exercise. However, there also exist patients who have muscle enlargement secondary to neuromuscular diseases [109]. Muscle atrophy always accompanies neurogenic disorders, but muscle enlargement, both hypertrophy and pseudohypertrophy, has also been found in neurogenic disorders [109]. In these neurogenic diseases, intact type II myofibers in partially denervated muscle probably experience hypertrophy due to the increased workload; stretching also results in the hypertrophy of type I myofibers, even when denervated [109,110]. However, this aberrant hypertrophy does not completely compensate for denervated atrophic muscle fibers in the same muscle. Hence, such enlarged muscles are paradoxically weak and are regarded as a pathological overgrowth.

Undoubtedly, the PI3K/Akt/mTOR signaling pathway has emerged as a pivotal regulator of glycolytic muscle growth and metabolism, and the over-expression of Akt accelerates skeletal muscle hypertrophy [111]. Highly expressed miR-199a-3p in skeletal muscle can target and inhibit IGF-1, mTOR and RPS6KA6 to activate the Akt/mTOR signaling pathway and partially block myoblast differentiation [112]. On the contrary, down-regulated miR-199a-3p can promote myoblast differentiation and myotube hypertrophy [112]. Similarly, miR-125b also regulates muscle hypertrophy by targeting IGF-2 to activate kinase-independent mTOR signaling in vitro and in vivo [113]. In addition, the myogenesis-associated lncRNA lnc-mg promotes myogenesis by functioning as a ceRNA for miR-125b to regulate IGF-2 [69]. Phenotypically, the knockout of lnc-mg in skeletal muscle leads to muscle atrophy and the loss of muscle endurance during exercise, while the over-expression of lnc-mg accelerates muscle hypertrophy [69]. In addition, an aging-related lncRNA Chronos was identified as an Akt inhibitor to suppress muscle hypertrophy [114].

**Table 2 cells-08-00988-t002:** Long non-coding RNAs (lncRNAs) in the myogenesis of skeletal muscle satellite cells.

lncRNA	Function	Partner	Reference
Sirt1 AS	Inhibits muscle formation	Sirt1; miR-34a	[27]
1/2-sbsRNA(B2)	Promotes cell activation	TRAF6	[60]
Uc.283+A	Promotes cell activation	pri-miR-195	[57]
H19	Maintains cell quiescence	-	[51]
	Promotes cell differentiation	MyHC; miR-675-3p; miR-675-5p	[53]
	Promotes cell differentiation	Sirt1; FoxO1	[54]
linc-MD1	Promotes cell differentiation	miR-133; miR-135	[64]
LncMyoD	Inhibits cell proliferation	IMP2	[68]
lnc-mg	Promotes cell differentiation	miR-125b	[69]
linc-YY1	Promotes cell differentiation	YY1	[70]
SMN-AS1	Promotes muscle atrophy	SMN; PRC2	[104]
Pvt1	Promotes muscle atrophy	c-Myc	[106]
lncISR1	Inhibits muscle atrophy	miR-15a/15b-5p/15c-5p	[107]
MUMA	Inhibits muscle atrophy	miR-762	[108]
Chronos	Inhibits muscle hypertrophy	Akt	[114]

## 4. Concluding Remarks

Here, we systematically summarized ncRNAs’ regulatory networks in the proliferation, differentiation and self-renewal of skeletal muscle satellite cells. Meanwhile, we also reviewed ncRNAs’ roles in muscle formation and muscle injury repair. With the unveiling of the mysterious functions of ncRNAs in skeletal muscles, research into their functions has become more and more in-depth, the mechanism and mode of action have gradually become more complex and the regulatory network tends to be clearer and more perfect, revealing that ncRNAs are an important part of satellite cells’ regulatory networks. While much progress has been made in identifying and verifying specific ncRNAs in skeletal muscle satellite cells, the relationship between ncRNAs and various muscle diseases has yet to be fully understood. Furthermore, ncRNAs’ function in myogenesis has provided important contributions for artificial organs in vitro and a new reference target for the treatment of muscle diseases [115,116], but how to harness these ncRNAs to develop effective and economic diagnostic and therapeutic tools is still a question to be further addressed in the future.

## Figures and Tables

**Figure 1 cells-08-00988-f001:**
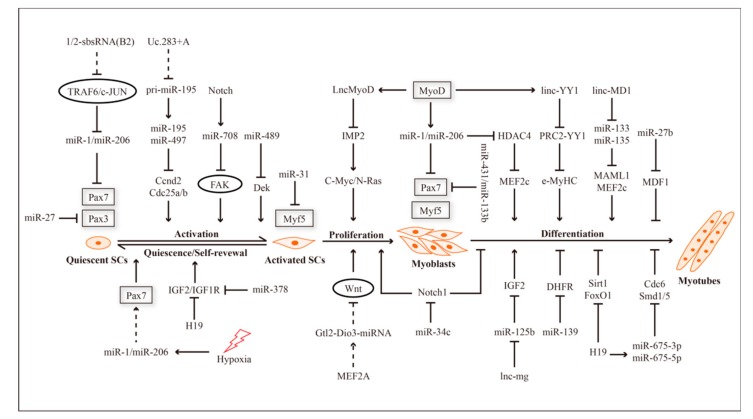
Non-coding RNA (ncRNA)-mediated regulatory networks in the myogenesis of skeletal muscle satellite cells.

**Figure 2 cells-08-00988-f002:**
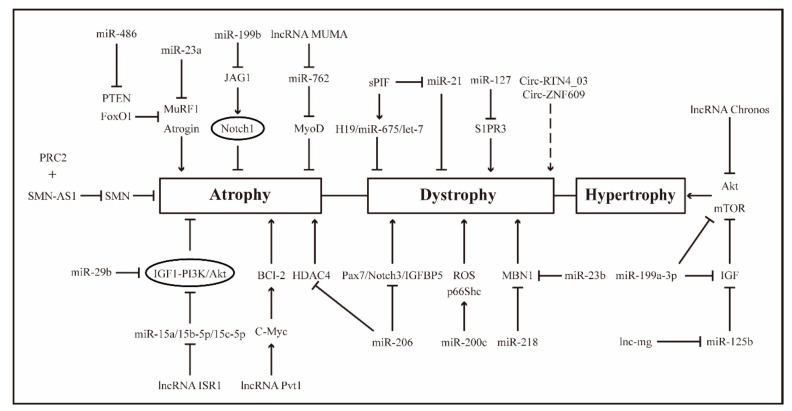
ncRNAs regulate muscle diseases such as dystrophy, atrophy and hypertrophy.

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
