# Peer review of "Non-Coding RNA Regulates the Myogenesis of Skeletal Muscle Satellite Cells, Injury Repair and Diseases"

_cells, 2019, doi:10.3390/cells8090988_

Round 1

Reviewer 1 Report

The authors have systematically reviewed the ncRNAs regulation networks in the proliferation, differentiation and self-renewal of skeletal muscle satellite cells. Moreover, they summarize the role of ncRNAs in muscle formation and muscle injury repair. The review expand the knowledge about skeletal muscle development and highlight targets for treatment of muscle dystrophy, atrophy and hypertrophy.   

Author Response

Dr. Editor

Thank you very much for your kind information regarding your comments and the reviewer’s criticisms for our manuscript. We have made a detailed reversion on the manuscript according to the referees’ criticisms and suggestions, and resubmit the revised manuscript for your consideration of publication in the Cells.

Yours sincerely

Shou-Long Deng, CAS Key Laboratory of Genome Sciences and Information, Beijing Institute of Genomics, Chinese Academy of Sciences, 100101 Beijing, China.

Comments and Suggestions for Authors

The authors have systematically reviewed the ncRNAs regulation networks in the proliferation, differentiation and self-renewal of skeletal muscle satellite cells. Moreover, they summarize the role of ncRNAs in muscle formation and muscle injury repair. The review expand the knowledge about skeletal muscle development and highlight targets for treatment of muscle dystrophy, atrophy and hypertrophy.  

Response: Thank you very much for your encouragement and support, we will continue work harder and thank you again for your approval.

Reviewer 2 Report

The present review represents a good picture of the current knowledge on the role of the ncRNA involved in the regulation of skeletal muscle satellite cell fate, but needs a major revision. The mechanism of action is not explained with sufficient details for many ncRNAs. Indeed, sometimes the role of a ncRNA is not explained in the same paragraph where the ncRNA is mentioned, leading to misunderstanding and confusion. Full name of molecules is often missing, but there are only abbreviations. Moreover, the text is difficult to understand since many sentences are poorly structured or even truncated.

It could be interesting to extend the study to other muscular diseases, adding for example a paragraph on the role of ncRNAs in rare muscular diseases.

Author Response

Dr. Editor

Thank you very much for your kind information regarding your comments and the reviewer’s criticisms for our manuscript. We have made a detailed reversion on the manuscript according to the referees’ criticisms and suggestions, and resubmit the revised manuscript for your consideration of publication in the Cells.

Yours sincerely

Shou-Long Deng, CAS Key Laboratory of Genome Sciences and Information, Beijing Institute of Genomics, Chinese Academy of Sciences, 100101 Beijing, China.

Comments and Suggestions for Authors

The present review represents a good picture of the current knowledge on the role of the ncRNA involved in the regulation of skeletal muscle satellite cell fate, but needs a major revision. The mechanism of action is not explained with sufficient details for many ncRNAs. Indeed, sometimes the role of a ncRNA is not explained in the same paragraph where the ncRNA is mentioned, leading to misunderstanding and confusion. Full name of molecules is often missing, but there are only abbreviations. Moreover, the text is difficult to understand since many sentences are poorly structured or even truncated.

It could be interesting to extend the study to other muscular diseases, adding for example a paragraph on the role of ncRNAs in rare muscular diseases.

Response: We are very grateful to the reviewer for this constructive comment. According to reviewer’s suggestion, we have complementally explained the mechanism of some ncRNAs and introduced the full name of molecules in the manuscript. In addition, we further expand the review of ncRNAs and skeletal muscle disease, and added the introduction about amyotrophic lateral sclerosis (ALS) and spinal muscular atrophy (SMA). Finally, the full text language has been checked by a native English-speaking colleague and professional English editing service to make it easier to understand.

Reviewer 3 Report

The manuscript by Zhao and colleagues aims at giving a review about the role of non-coding RNAs in skeletal muscle stem cells, that are also known as satellite cells. This is an interesting and timely topic for our general understanding of gene regulation during development, and for the elucidation of molecular mechanisms in muscle biology. While the authors review several key findings that have been made in the past, the manuscript unfortunately also has several drawbacks. One problem throughout the text is the poor English, which makes it often difficult to understand the meaning of the sentences. In addition, division of part 2 into subparts causes a lot of redundancies as the steps of satellite cell development are overlapping and thus the functions of the ncRNAs cannot be clearly separated. Part 2 shows also a big overlap with the reference #55, a recent review by Goncalves and Armand, Noncoding RNA Res, 2017. Therefore, I would suggest to rather expand part 3 of the manuscript and include the information of part 2 there. My further specific concerns are:

page 1, line 23:

“skeletal satellite cells”

The word “muscle” is missing!

page 1, line 25 and p. 2, l. 75:

How can a review “expand the understanding” of a research field? The authors should rephrase these sentences.

p. 1, l. 27 and p. 2, l. 77:

Why do the authors consider muscle hypertrophy a disease?

p.1, whole text:

The word “skeletal” is used too excessively.

p. 3, Table 2:

The list is very short. Even in the manuscript several other lncRNAs are mentioned, e.g. Sirt1 AS, Pvt1 and Chronos.

p. 3, l. 10:

 “post-transcriptional translation“

Translation takes always place post-transcriptionally!

p. 4, l. 116:

“lncRNA Uc.283+A can rapidly destroy the functional miR-195” Is the lncRNA sequestering the miRNA, or degrading it?

p. 4, l. 118:

Here and throughout the text acronyms for proteins and genes like IGF1R should be introduced.

p.4/5, l. 153/154:

“This discrepancy may be due to the inhibition of both miR-1 and miR-206 by antisense oligonucleotides.”

The sentence is out of context.

p. 5, l. 190:

“Linc-MD1 is localized in the cytoplasm and is polyadenylated acts as a natural decoy for miRNAs to involve in the timing of muscle differentiation [60].”

“and” before “acts” is missing!

p. 5, l. 193-197:

“by sponges”, “Mechanically”, “lncMyoD directly binding ... and results”

Wrong grammar or word usage!

p. 5, l. 202:

“Linc-YY1, which is also induced by MyoD, interacts with YY1 gene”

The lncRNA interacts with YY1 protein, not the gene.

p. 6

whole part 2.3 is redundant with the previous parts

p. 6, l. 234

“qui9escence” ??

p. 7, l. 265/266

“In detail, miR-200c might increase the ROS production and phosphorylates p66Shc in Ser-36 to cause the muscle dystrophy and wasting [91].”

How can a miRNA phosphorylate a protein?

p. 7, l. 267/268

“The latest research showed that sPIF can promote myoblast differentiation and utrophin expression while suppress the fibrosis in DMD via lncRNA H19/miR-675/let-7 and miR-21 pathways [92].”

The context is not clear. What is sPIF, how does utrophin expression play a role here?

p. 8, part 3.3

How can miRNAs activate the PI3K/Akt/mTOR pathway by targeting its factors?

p. 9, l. 321

“increasing evidence of publications”

Evidence for what?

p. 9, l. 328

“ncRNAs function in myogenesis has provided important contributions for artificial organs in vitro”

This statement needs to be supported by a citation.  

Author Response

Dr. Editor

Thank you very much for your kind information regarding your comments and the reviewer’s criticisms for our manuscript. We have made a detailed reversion on the manuscript according to the referees’ criticisms and suggestions, and resubmit the revised manuscript for your consideration of publication in the Cells.

Yours sincerely

Shou-Long Deng, CAS Key Laboratory of Genome Sciences and Information, Beijing Institute of Genomics, Chinese Academy of Sciences, 100101 Beijing, China.

Comments and Suggestions for Authors

The manuscript by Zhao and colleagues aims at giving a review about the role of non-coding RNAs in skeletal muscle stem cells, that are also known as satellite cells. This is an interesting and timely topic for our general understanding of gene regulation during development, and for the elucidation of molecular mechanisms in muscle biology. While the authors review several key findings that have been made in the past, the manuscript unfortunately also has several drawbacks. One problem throughout the text is the poor English, which makes it often difficult to understand the meaning of the sentences. In addition, division of part 2 into subparts causes a lot of redundancies as the steps of satellite cell development are overlapping and thus the functions of the ncRNAs cannot be clearly separated. Part 2 shows also a big overlap with the reference #55, a recent review by Goncalves and Armand, Noncoding RNA Res, 2017. Therefore, I would suggest to rather expand part 3 of the manuscript and include the information of part 2 there. My further specific concerns are:

Response: We are very grateful to the reviewer for this constructive comment. According to reviewer’s suggestion, we have readjusted the layout of the part 2 and expanded the content of the part 3. Finally, the full text language has been checked by a native English-speaking colleague and professional English editing service to make it easier to understand.

page 1, line 23:

“skeletal satellite cells”

The word “muscle” is missing!

Response: Thanks to reviewer for your careful review. The word “muscle’’ has added.

page 1, line 25 and p. 2, l. 75:

How can a review “expand the understanding” of a research field? The authors should rephrase these sentences.

Response: Thanks for your careful review. We have revised the unreasonable wording by “…help us deeply understand the …”.

1, l. 27 and p. 2, l. 77:

Why do the authors consider muscle hypertrophy a disease?

Response: Thanks to reviewer for your review. Healthy muscles enlarge through exercise. However, there are also exist patients who have muscle enlargement secondary to neuromuscular diseases. In these neurogenic disease, intact type II myofiber in partially denervated muscle probably hypertrophy due to the increased workload; And stretch also results type I myofiber hypertrophy even when denervated. However, this hypertrophy does not completely compensate for denervated atrophic muscle fibers in the same muscle. Hence, such enlarged muscles are paradoxically weak and is regards as a pathological overgrowth. We have added this reason into the manuscript.

p.1, whole text:

The word “skeletal” is used too excessively.

Response: Thanks to reviewer’s quite valuable suggestion. We have appropriately deleted some word of “skeletal”.

3, Table 2:

The list is very short. Even in the manuscript several other lncRNAs are mentioned, e.g. Sirt1 AS, Pvt1 and Chronos.

Response: Considering reviewer’s suggestion, we have added some lncRNAs in Table 2.

3, l. 10:

“post-transcriptional translation“

Translation takes always place post-transcriptionally!

Response: We are very sorry for our vaguely description. We have deleted the word “post-transcriptional”.

4, l. 116:

“lncRNA Uc.283+A can rapidly destroy the functional miR-195” Is the lncRNA sequestering the miRNA, or degrading it?

Response: Thanks for your constructive comments. lncRNA Uc.283+A blocks the formation of mature miR-195 by controlling the processing of pri-miR-195. We have explained it in the manuscript.

4, l. 118:

Here and throughout the text acronyms for proteins and genes like IGF1R should be introduced.

Response: We are sorry for our neglect. We have introduced all the acronyms for proteins and genes in the manuscript.

p.4/5, l. 153/154:

“This discrepancy may be due to the inhibition of both miR-1 and miR-206 by antisense oligonucleotides.”

The sentence is out of context.

Response: We are very sorry for our vaguely description. We have deleted this illogical sentence.

5, l. 190:

“Linc-MD1 is localized in the cytoplasm and is polyadenylated acts as a natural decoy for miRNAs to involve in the timing of muscle differentiation [60].”

“and” before “acts” is missing!

Response: Thanks for your careful review. We have added the word “and’’ before “acts’’.

5, l. 193-197:

“by sponges”, “Mechanically”, “lncMyoD directly binding ... and results”

Wrong grammar or word usage!

Response: Thanks for your careful review. We have corrected the wrong grammar and word usage in the manuscript.

5, l. 202:

“Linc-YY1, which is also induced by MyoD, interacts with YY1 gene”

The lncRNA interacts with YY1 protein, not the gene.

Response: Thanks for your careful review. The word “gene’’ was modified to “protein”.

6

whole part 2.3 is redundant with the previous parts

Response: Considering reviewer’s suggestion, we have reorganized the part 2.

6, l. 234

“qui9escence” ??

Response: Thanks to reviewer for your careful review. The word “qui9escence” was modified to “quiescence”.

7, l. 265/266

“In detail, miR-200c might increase the ROS production and phosphorylates p66Shc in Ser-36 to cause the muscle dystrophy and wasting [91].”

How can a miRNA phosphorylate a protein?

Response: We are very sorry for our vaguely description. In fact, the up-regulated miR-200c decreases NO, increases ROS production, and induces p66Shc protein phosphorylation in Ser-36. Therefore, a miRNA cannot directly phosphorylate a protein. We have updated the description in the manuscript to make it easier to understand.

7, l. 267/268

“The latest research showed that sPIF can promote myoblast differentiation and utrophin expression while suppress the fibrosis in DMD via lncRNA H19/miR-675/let-7 and miR-21 pathways [92].”

The context is not clear. What is sPIF, how does utrophin expression play a role here?

Response: We are very sorry for our vaguely description. Preimplantation factor (PIF) is an evolutionary conserved 15-amino acid peptide secreted by viable mammalian embryos. Synthetic PIF (sPIF) reproduces the protective/regenerative effects of the endogenous peptide in immune disorders and transplantation models. We have added the detailed description of sPIF in the manuscript. In addition, sPIF can promote DMD myoblasts differentiation, increase the expression of utrophin, and reduce muscle fibrosis possibly via the upregulation of miR-675 and inhibition of miR-21 expression. We also have updated this description to make it easier to understand.

8, part 3.3

How can miRNAs activate the PI3K/Akt/mTOR pathway by targeting its factors?

Response: Thanks to reviewer for your careful review. In fact, miRNAs inhibit the expression of target genes, then, down-regulated target genes induce the activation of PI3K/Akt/mTOR pathway. We have modified this description in the manuscript.

9, l. 321

“increasing evidence of publications”

Evidence for what?

Response: We are very sorry for our vaguely description. We have updated this description in the manuscript.

9, l. 328

“ncRNAs function in myogenesis has provided important contributions for artificial organs in vitro”

This statement needs to be supported by a citation.

Response: Thanks for your constructive comments. We have modified our statement to “ncRNAs function in myogenesis would provide an important contribution for artificial organs in vitro”. Actually, the evidence has shown that ncRNAs has provided important contribution for artificial organs in vitro. For example, joint inhibition of miR-133a and miR-696 accelerated cell differentiation, elevated the metabolic coactivator PGC-1α, and increased the contractile force in 3D engineered human skeletal muscle bundles [1]. And another study demonstrated that miRNA mediation has a downstream functional effect on tissue-engineered constructs. In detail, differentiation of skeletal myoblasts in vitro may be enhanced by transient transfections of miRNAs. Peak forces exhibited by anti-miR-133 BAMs were on average 20% higher than its corresponding negative control to electrical stimulation, and responses to electrical stimulation in miRNA-mediated BAMs (along with negative controls) were similar to nontransfected controls. Immunostaining also showed more distinct striations and myofiber organization in BAMs with miR-133 inhibition over the negative control. Fiber diameters were also significantly larger in these BAMs over both the nontransfected and negative controls [2].

[1] Cheng, C. S.; Ran, L.; Bursac, N.; Kraus, W. E.; Truskey, G. A., Cell Density and Joint microRNA-133a and microRNA-696 Inhibition Enhance Differentiation and Contractile Function of Engineered Human Skeletal Muscle Tissues. Tissue Eng Part A 2016, 22 (7-8), 573-83.

[2] Rhim, C.; Cheng, C. S.; Kraus, W. E.; Truskey, G. A., Effect of microRNA modulation on bioartificial muscle function. Tissue Eng Part A 2010, 16 (12), 3589-97.

Round 2

Reviewer 3 Report

I appreciate the efforts made by the authors to improve the manuscript. All my previous concerns have been mostly satisfactorily addressed. However, I would still suggest to soften the statement in the abstract "This work will help us to deeply understand skeletal muscle biology...", as a review can just summarize existing data. I would also suggest to use the term "aberrant hypertrophy" or similar for the pathological condition the authors refer to. Lastly, the 2 publications mentioned in the authors response should also be cited in the manuscript.

[1] Cheng, C. S.; Ran, L.; Bursac, N.; Kraus, W. E.; Truskey, G. A., Cell Density and Joint microRNA-133a and microRNA-696 Inhibition Enhance Differentiation and Contractile Function of Engineered Human Skeletal Muscle Tissues. Tissue Eng Part A 2016, 22 (7-8), 573-83.

[2] Rhim, C.; Cheng, C. S.; Kraus, W. E.; Truskey, G. A., Effect of microRNA modulation on bioartificial muscle function. Tissue Eng Part A 2010, 16 (12), 3589-97.

Author Response

Response: We are very grateful to the reviewer for this constructive comment. According to reviewers suggestion, we have modified the statement in the abstract "This work will help us to deeply understand skeletal muscle biology..." by "This work will systematically summarize the role of ncRNAs in myogenesis… ". We also use term "aberrant hypertrophy" and emphasize it in the manuscript. And, the 2 publications mentioned have been cited. Lastly, due to our carelessness, we forgot to modify the references in Table 1 and Table 2 at the first revision. Here, we have updated the references in Table 1 and Table 2. Thank you again for your suggestion.